# Exclusive Breastfeeding Protects Young Children from Stunting in a Low-Income Population: A Study from Eastern Indonesia

**DOI:** 10.3390/nu13124264

**Published:** 2021-11-26

**Authors:** Hamam Hadi, Fatimatasari Fatimatasari, Winda Irwanti, Chahya Kusuma, Ratih Devi Alfiana, M. Ischaq Nabil Asshiddiqi, Sigit Nugroho, Emma Clare Lewis, Joel Gittelsohn

**Affiliations:** 1Alma Ata Graduate School of Public Health, the University of Alma Ata, Yogyakarta 55183, Indonesia; chahyakusuma@almaata.ac.id (C.K.); sigitnugroho@almaata.ac.id (S.N.); 2Alma Ata Center for Healthy Life and Foods (ACHEAF), the University of Alma Ata, Yogyakarta 55183, Indonesia; 3Department of Midwifery, Faculty of Health Sciences, the University of Alma Ata, Yogyakarta 55183, Indonesia; fatimatasari@almaata.ac.id (F.F.); ratihdevi@almaata.ac.id (R.D.A.); 4Department of Nutrition, Faculty of Health Sciences, The University of Alma Ata, Yogyakarta 55183, Indonesia; windairwanti@almaata.ac.id; 5Department of Nursing, Faculty of Health Sciences, the University of Alma Ata, Yogyakarta 55183, Indonesia; ischaq.nabil@almaata.ac.id; 6Center for Human Nutrition, Department of International Health, Johns Hopkins Bloomberg School of Public Health, 615 N. Wolfe Street, Baltimore, MD 21205, USA; elewis40@jhu.edu (E.C.L.); jgittel1@jhu.edu (J.G.)

**Keywords:** exclusive breastfeeding, children, stunting, low-income, Indonesia

## Abstract

The prevalence of stunting in young Indonesian children is the highest among countries belonging to the Association of Southeast Asian Nations (ASEAN). Breastfed children are reported to grow better than non-breastfed. The present study examined the protective effect of exclusive breastfeeding against stunting in children under two years old (CU2) and its interaction with monthly household expenditure. Secondary analyses were conducted based on a 2012 cross-sectional study including 408 children aged 6–24 months and their caregivers from 14 villages in rural Eastern Indonesia. Data on breastfeeding history, childcare, and household expenditures were collected using structured questionnaires. Focus Group Discussions (FGDs) were conducted in each village (n = 14). Nearly two-thirds (61%) of caregivers who identified as the biological mother exclusively breastfed their child at 6 months. Exclusively-breastfed CU2 from poorer households were 20% less likely to be stunted than their non-exclusively-breastfed peers. Further, exclusively-breastfed CU2 from wealthier households were 50% less likely to be stunted than non-exclusively-breastfed CU2 from poorer households. FGDs revealed that some mothers were unaware of the importance of recommended breastfeeding practices. Exclusive breastfeeding may protect low-income children against stunting. Health promotion to improve caregiver motivation to exclusively breastfeed is critical in the present setting and beyond.

## 1. Introduction

Child undernutrition, especially stunting, remains a global public health challenge in the 21st century [1]. The prevalence of stunting is concerningly high in Indonesia and East Nusa Tenggara (NTT). Nationally, based on the Basic Health Research report in 2018, the prevalence of stunting among children under 5 years of age in Indonesia is 30.8%, while in NTT Province, the prevalence is 42.6% [2]. Stunting is defined as having a height-for-age below two standard deviations of the WHO Child Growth Standards median [3] and is a consequence of inadequate nutrition and/or recurrent infections or chronic diseases which cause poor nutrient intake, absorption, and utilization [4]. Stunting causes short- and long-term consequences, including impaired physical and cognitive development, decreased productivity and health status, and increased risk of degenerative diseases in adulthood [5].

To reduce stunting prevalence more effectively, rigorous prevention strategies need to be defined and implemented properly. Exclusive breastfeeding has been reported to be effective for maintaining optimal growth of young children [6]. Although several studies found that formula feeding can lead to significantly greater weight gain [7,8,9,10] and length gain [7,11] compared to breastfeeding, this weight gain is often considered to occur too quickly and can result in an infant being overweight [7,12]. Furthermore, the use of formula feeding is not recommended in low- and middle-income country (LMIC) settings due to poor sanitation and the subsequent risk of contamination of the water with microbes and toxins used for the preparation of formula [13]. Therefore, the impact of exclusive breastfeeding in protecting young children from stunting in remote, diverse economic environments requires further investigation and promotion.

The prevalence of women who exclusively breastfeed their infants to 6 months of age remains low, with only 41% globally and 37% in LMICs [14,15]. In Indonesia, breastfeeding is widely practiced, but the prevalence of exclusive breastfeeding at 6 months is only 37.3% [2] and has not improved despite recommendations to exclusively breastfeed made by the World Health Organization’s Infant and Young Child Feeding (WHO IYCF) guidelines. Breastfeeding has been reported to be more prevalent among poor communities in Indonesia, but very few of these communities promote exclusive breastfeeding [16]. 

Poverty has been widely recognized as a proxy of stunting [17,18,19]. Previous studies have indicated a dose-response relationship between monthly household expenditure and stunting, such that the lower the monthly household expenditure, the more likely children are to be stunted; and vice versa [2,20,21]. Yet, the relationship may differ depending on the existence of effect modifiers such as exclusive breastfeeding. We hypothesize that young children from poor households could be protected from stunting if they are exclusively breastfed. If evidence is found in support of this hypothesis, we would expect that improved exclusive breastfeeding could serve as an effective strategy for implementation by the Indonesian government, which has recently declared a strong commitment to accelerate the reduction of stunting prevalence up to 14% by the end of 2024 [22]. The promotion of exclusive breastfeeding could also be adopted as an effective strategy to reduce stunting prevalence in other low income countries where exclusive breastfeeding is locally and culturally acceptable. 

The present analysis sought to examine the role of exclusive breastfeeding in reducing stunting prevalence among CU2 attributable to low household expenditures in a low-income Indonesian population. We also sought to understand local perceptions related to these behaviors and influential factors in order to design culturally appropriate health promotion strategies.

## 2. Materials and Methods

### 2.1. Study Population and Sampling Methods

The data set used in this analysis was a part of the study on “behavioral analysis and food consumption/dietary practices amongst CU2, elementary school age children, pregnant and lactating women in the Timor Tengah Selatan district of Nusa Tenggara Timur province, Indonesia”, conducted in 2012. In the present analysis, we used cross-sectional data of CU2 to particularly examine breastfeeding, monthly household expenditure, and their interaction as determinants of stunting in this low-income population.

Subjects were caregivers of children aged 6–24 months living in the Timor Tengah Selatan (TTS) district, where the prevalence of household food insecurity [23,24] and stunting [25] are both high. More specifically, subjects were recruited from the West Amanuban and Kie Districts of TTS, each consisting of 7 villages (14 villages total). Kie was selected because it had the lowest underweight prevalence in TTS based on the 2010 District Health Profile [23], while West Amanuban was selected because it had the highest prevalence of undernutrition in TTS and was categorized as an area with high food insecurity. Seven villages from each district were then chosen based on study accessibility (i.e., condition of roads, etc.) which was determined in accordance with district government officials. In each village, approximately 30 caregivers of CU2 were randomly selected, resulting in a total sample of 408 caregivers. 

In addition, Focus Group Discussions (FGDs) were conducted in each village (n = 14), and each FGD included 10–12 caregivers. Caregivers were recruited to participate in the FGD if they lived in either the West Amanuban or Kie Districts for at least one year prior to the study, and if they provided informed consent to participate.

### 2.2. Data Collection

Quantitative survey data were collected on breastfeeding behaviors, demographic and socioeconomic factors including monthly household expenditure, via structured questionnaires administered by trained interviewers (i.e., Diploma of Nutrition Science graduates). The length of CU2 was measured by trained enumerators using a recumbent length board with a precision of 0.1 cm. Length-for-age Z-scores were computed using the WHO Anthro 2005 criteria based on the 2005 WHO Standard Growth Reference for CU2. FGDs were conducted by trained data collectors and translators with caregivers to examine perceptions, beliefs, barriers, and enablers related to infant feeding practices. Translation was occasionally necessary given that most participants spoke a local language (Bahasa Timor or tribal language) different from the national language.

### 2.3. Variables 

Being categorized as stunted referred to having a length-for-age z-score < −2, whereas being categorized as not stunted referred to having a length-for-age z-score ≥ −2. We used a combination of two questions to determine whether subjects practiced exclusive breastfeeding: (1) “Did your child only get breastmilk for the first six months after birth?”, and (2) “Until what age did your child receive only breastmilk?” Exclusive breastfeeding was defined as a positive response to the first question and at least six months to the second question. 

Household expenditures were collected as monthly expenditures on food and non-food items. Respondents were asked about how much they spend for rice, corn, starches, meat and fish, vegetable, snacks, and other food expenses in a month. For non-food items, they were asked about expenditures for electricity, schools, social activities/parties/gifts for special occasions, clothes, health, cigarettes, betel nut, and others. We summed all expenses then categorized them into two categories, below and equal to or above the regional cut off minimum wage in the NTT Province (IDR 850,000 or approximately ± USD 60.00). In general, socioeconomic status is usually measured using household income [26,27,28]. However, in practice, household income is often difficult to accurately obtain due to the reluctance of respondents to state their income, as has been observed in previous studies conducted among the Indonesian population. The measurement of socioeconomic status using monthly household expenditure has been found to be effective and reliable in past research [29]. Therefore, instead of household income we chose to use monthly household expenditure to measure socioeconomic status in the present study.

Mother’s age was categorized into three categories; under 20 years, 20–34 years, or 35 years or older. Birth order refers to the order in which the target child was born into his/her family. For all analyses, we categorized birth order: into number 1, 2, 3, 4, or 5 or above.

Caregiver education was categorized into five levels (uneducated, graduated elementary school, graduated junior high school, graduated senior high school, and graduated university) for descriptive univariate analyses, while for bivariate analyses we further simplified these categories into four levels (uneducated, graduated elementary school, graduated junior high school, graduated senior high school or higher).

In our descriptive analyses, we described caregiver occupation using four categories, namely, farmer/animal husbandry/fisherman/laborer, civil servant/police/military/entrepreneur, housewife/unemployed, or other.

### 2.4. Data Analysis

#### 2.4.1. Survey Data

Chi-square tests were performed to assess demographic differences between children who were exclusively breastfed and children who were not exclusively breastfed. Bivariate and multivariate logistic regression models were then used to examine the associations between stunting and household expenditure and their interaction with exclusive breastfeeding, adjusting for demographic and socioeconomic factors. All factors which had a *p*-value of <0.25 in the bivariate analysis were included in the model for multivariate analysis. All data analyses were conducted using STATA v.15 MP [30].

#### 2.4.2. Focus Group Data

FGDs were led using a structured questionnaire guide. They were recorded by facilitators, and as soon as the FGDs were finished, facilitators transcribed the audio recording into a script for each meeting. We then analyzed each script and reported key themes and patterns.

## 3. Results

### 3.1. Survey Sample Characteristics

Four-hundred and eight caregivers of children 6–24 months of age were interviewed. Approximately half of the target children were male, and the average age was 13.6 months (standard deviation (SD) = 5.2 months). Most children (61.8%) were between 12 to 24 months, and were classified as the first (28.9%), second (21.6%) or third (20.6%) child born into the family.

Mothers reported having between 1–10 children with most reporting having one (27.9%), two (22.5%), or three (19.6%) children. Most mothers (70.1%) were 20 to 35 years of age with an elementary school (45.4%) or junior high school (24.3%) level education. The father’s educational backgrounds were similar to the mother’s; most graduated from elementary school (35.5%) and junior high school (27.4%). Most mothers (84.8%) reported not working outside of the home, while most fathers (79.5%) worked as farmers, ranchers, or fishermen. More than 80% of households in this study reported an expenditure below the Regional Minimum Wage in NTT (2012) of IDR 850,000 (±USD 60.00) per month (Table 1). 

### 3.2. Factors Associated with CU2 Stunting

We found that 44.1% of CU2 in our sample were stunted, 13.2% of which were severely stunted. Children who were boys, and whose mothers had a lower educational attainment, were more likely to be stunted compared to children who were girls and whose mothers had a higher educational attainment (Table 2). Younger children and children who were primarily cared for by their biological mother were less likely to be stunted than older children and children who were not primarily cared for by their biological mother. In addition, children from households with a monthly income of less than the regional minimum wage were more likely to be stunted compared to those from households with a monthly income of greater than or equal to the regional minimum wage (Table 2). 

### 3.3. The Prevalence of Stunting by Monthly Household Expenditure, and the Interaction between Exclusive Breastfeeding and Monthly Household Expenditure 

We found no significant direct relationship between exclusive breastfeeding and stunting in this sample, but we found a significant relationship between monthly household expenditures and exclusive breastfeeding (Table 3) Further analyses were conducted to explore the relationship between stunting prevalence, exclusive breastfeeding, and monthly household expenditures. Interestingly, we found a reverse U-curve when stunting prevalence was divided according to monthly household expenditure quintiles. The prevalence of stunting in the 1st quintile was relatively low (39.5%), then increased in the 2nd, 3rd and 4th quintiles (51,1%, 52,4%, and 42,7%, respectively) and finally decreased to 34.6% in the 5th quintile (Figure 1).

To further explore this finding, a new variable (interaction term between exclusive breastfeeding and monthly household expenditure) was created. In addition to this interaction term, other variables associated with stunting with a *p* < 0.1 were included in a multiple logistic regression model. Backward stepwise regression was used to create a model that met the criteria of ‘Goodness of Fit’ as displayed in Figure 2. 

We found that compared to CU2 with a monthly household expenditure of less than the mean who were not exclusively breastfed, CU2 with a monthly household expenditure of less than the mean were 20% less likely to be stunted if they were exclusively breastfed. Further, CU2 with a monthly household expenditure greater than or equal to the mean but who were not exclusively breastfed were 40% less likely to be stunted than their peers. In contrast, CU2 with a monthly household expenditure of greater than or equal to the mean and who were exclusively breastfed were 50% less likely to be stunted than their peers (Figure 2). 

We also conducted a qualitative component of the present study to complement the quantitative secondary analysis and to add richness to the context of our research questions, namely in the form of FGDs among breastfeeding mothers. Results indicate that most breastfeeding mothers knew that exclusive breastfeeding referred to giving only breastmilk until six-months-old and some of them also practiced this. However, some mothers reported giving food other than breastmilk to babies under six months for various reasons including: not producing enough breastmilk, not having a place to breastfeed in the work place, being busy, and children frequently crying. Some lactating mothers stated that they fed their children foods other than breastmilk as early as 2–3 months of age. Among these mothers, formula milk and rice porridge were the most commonly fed items. 

Most lactating mothers stated that they had received information about and encouragement for exclusive breastfeeding from midwives at the community health centre (*Puskesmas*) and cadres at integrated health centres in their villages (*Posyandu*). Mothers were told to practice exclusive breastfeeding for six months, but most mothers stated that no explanation was given regarding why this was necessary. Cadres are residents who do not receive formal health education, but have been given health training by the community health center to be able to provide basic health education to villagers. However, based on in-depth interviews with several cadres, cadres also often had an inappropriate understanding of the nutritional status of stunted children.

The communication channel between cadres and mothers happened face-to-face during antenatal care (ANC) visits to Puskesmas or Posyandu. At the time the present study took place, there were no alternative options for mothers to obtain this education other than to meet cadres in-person, mostly due to a lack of available smartphones and internet connectivity, as well as a non-optimal power grid in the study region. Midwives and cadres meet with mothers most frequently throughout pregnancy and are therefore important figures for providing adequate information on postnatal care and exclusive breastfeeding. 

Our study found that 90.17% of respondents stated that mothers had given colostrum to their babies at birth. Mothers who did not give colostrum had various reasons, including fear of children getting sick because the color of the breastmilk was yellow (perceived to be ”stale”), having sore nipples, following generational tradition, their baby having diarrhea, their baby’s stomach being bloated, and in cases where the biological mother died during childbirth. These findings were supported by our FGD results, demonstrating the presence of several key cultural beliefs that influenced exclusive breastfeeding practices and colostrum feeding in the study region. Misconceptions about colostrum led to some mothers throwing it away, especially those who gave birth at home without the presence of a health worker. 

We also found that different regulations regarding birth care in several villages may influence different stunting rates across villages. For example, in some villages in Kecamatan Kie, mothers who deliver their babies not at the health center and/or without the assistance of health workers are fined IDR 500,000 (approximately USD 35.00). Mothers who live in these villages stated in FGDs that they obey the rule because they cannot afford to pay the fine. In doing so, mothers are more likely to give birth in a health center, which in turn may influence them to better comply with health recommendations including breastfeeding initiation and exclusive breastfeeding. The relationship between these context-specific factors and stunting should be further examined in future studies.

## 4. Discussion

To our knowledge, this is the first study to examine the interaction between exclusive breastfeeding and monthly household expenditure in association with stunting among CU2 in a rural Indonesian population. This study has supported our hypothesis that young children from poor households could be protected from stunting if they are exclusively breastfed. We also found that stunting was significantly associated with monthly household expenditure, age, primary caregiver type, and mother’s occupation. More importantly, in a poor setting where breastfeeding is commonly practiced, stunting was associated with monthly household expenditure in a reversed U-shape as opposed to in a linear manner. We found that the likelihood of being stunted was similar for children in the 1st quintile of monthly household expenditure compared to children in the 4th and the 5th quintiles. In contrast, children in the 2nd and 3rd quintiles were about twice as likely to be stunted than children from all other quintiles. Importantly, we also found that exclusively breastfed children were 25% less likely to be stunted than those who were not exclusively breastfed, although this association did not reach statistical significance. Breastfeeding was not directly associated with stunting, but it significantly reduced the odds of being stunted in association with lower monthly household expenditure.

It may be that the high prevalence of exclusive breastfeeding in the 1st quintile of monthly household expenditures protects CU2 from stunting, while CU2 from the 5th quintile of monthly household expenditures have greater access to nutritious foods and drinks. We previously reported that exclusive breastfeeding was more prevalent among poorer households [31]. In this sample, 70.4% of CU2 in the 1st quintile of monthly household expenditure were exclusively breastfed, but this prevalence declined to only 48.2% among CU2 in the 5th quintile. The mean household expenditure spent on food in this population was 36.6% ± 18.1% (mean ± SD). The proportion of food expense rose with increased monthly household expenditure. Every IDR 100,000 (USD 7.06) increase in total household expenditure was associated with an IDR 19,000 (USD 1.30) increase in food expenditures (*p* < 0.001). Additionally, every IDR 100,000 (USD 7.06) increase in household expenditure was followed by a 0.28 g/day increase in CU2 protein intake (*p* < 0.01). These findings are in accordance with other previous studies [32,33].

The latter finding is in agreement with Cortes and colleagues (2018), in which the authors found that the proportion of stunting was twice as large among children who were not breastfed compared to those who were breastfed [34]. A similar study also found that the mean length-for-age Z-score in exclusively breastfed infants was significantly higher than in those who were not exclusively breastfed [35]. This can also be described that giving complementary foods too early will increase the risk of stunting in children [36]. Another study found otherwise, reporting no significant difference in the proportion of stunting between children who were exclusively breastfed and those who were not exclusively breastfed. However, this study also found that infants who were exclusively breastfed were more resistant against infections, so the practice of exclusive breastfeeding may still play a crucial role in preventing stunting in that study sample, as stunting can result from repeated infections [37].

Among our study sample, children who were primarily cared for by their biological mother were 2.5 times less likely to be stunted. Similarly, Reyes and colleagues (2004) found that children who were primarily cared for by their biological mother were 2.2 times less likely to be stunted compared to children who were cared for by others [38]. One reason for this might be that children cared for by their biological mothers are more likely to be exclusively breastfed than those cared for by other household members [32,39]. We previously reported that Indonesian children cared for by their biological mothers were 4.6 times more likely to exclusively breastfed than those cared by others [32]. In the present study, the most common reasons given for feeding other foods besides breastmilk to CU2 were (1) caregiver perceptions of child hunger and fussiness, and (2) that the mother was not always available to breastfeed. These findings are similar to data collected in 2018 in the province of Eastern Nusa Tenggara, where common reasons for non-exclusive breastfeeding included (1) having insufficient milk (59.1%), (2) children being separated from their mothers (9.7%), and (3) medical reasons (8.4%) (2). Similarly, another study found that 93.2% of mothers reported insufficient milk production as a reason for cessation of exclusive breastfeeding [40]. Other studies reporting on factors inhibiting exclusive breastfeeding included insufficient knowledge, nipple wounds, receiving formula milk samples at the beginning of birth, mothers working outside of the home, mothers being sick, and a lack of support from family members to help provide exclusive breastfeeding [41,42]. 

FGDs revealed that some mothers in our study were unaware of the reasons for promotion of exclusive breastfeeding to six months of age, that they misunderstand the importance of colostrum, and that they did not understand proper breastfeeding practices when caring for a sick baby, or for themselves when sick. However, research has found that it is perfectly healthy to breastfeed a baby even when the mother is sick [43,44,45,46,47], with few medical exceptions [48,49].

This indicates that mothers in the study region may carry out health practices based on the instructions of health workers, but may not understand the importance of carrying out these practices. We suspect that this may be due to a lack of two-way communication between health workers and mothers, however, we were unable to further explore this hypothesis as we did not conduct interviews with midwives regarding the delivery of information to pregnant and post-partum mothers. Importantly, based on our interviews, lactating mothers receive most of their information passed down within their families from generation-to-generation. 

It is essential that health workers provide education to mothers during early ANC visits to ensure that mothers understand the importance of exclusive breastfeeding as a part of the strategy to reduce stunting. Previous studies have demonstrated that antenatal breastfeeding education is an effective way to increase the level of breastfeeding self-efficacy, which in turn increases exclusive breastfeeding practices [50,51,52] and positively changes breastfeeding behavior [53,54]. In addition, valuing the benefits of exclusive breastfeeding during pregnancy is a strong independent predictor of actual exclusive breastfeeding duration [55,56,57]. Mass media can be used as an effective tool for breastfeeding promotion; however, mass media is most effective when combined with other intervention strategies such as policy, advocacy, interpersonal communication, and community mobilization. This is largely due to the fact that informational and emotional support provided by partners, families, and health workers remain the biggest influence for mothers to breastfeed [58,59,60,61,62,63].

A few strengths and limitations of the present study should be noted. First, a strength of this study was its large sample size, which allowed us to look at the individual- and combined-effects of multiple risk factors on childhood stunting. The use of mixed methods is an additional strength of the present study, as the qualitative data provided rich context for understanding the quantitative data. The cross-sectional design of the present study was a limitation, as it restricted our ability to assess causation. A preferable design would have been a cohort study to assess the individual and combined effects of breastfeeding and monthly household expenditure on childhood stunting, by taking into account the sociodemographic status and different built environments of individuals. Future research should consider employing a cohort study design to examine these factors of interest. The fact that the data set used in this analysis were collected in 2012 could be another limitation of this study, as things might have changed. However, the prevalence of stunting among under five children in this province did not change much, as demonstrated in the recent National Health survey, which was still 42.6% [2]. In addition, the natural relationship between stunting and household expenditure as well as their interaction with breastfeeding might not change with the passage of time. 

## 5. Conclusions

In low-income populations where breastfeeding is common, exclusive breastfeeding in particular plays a crucial role in protecting young children from being stunted. Improving exclusive breastfeeding could be one of the cost-effective strategies to accelerate stunting reduction in Indonesia and other LMICs where exclusive breastfeeding is locally and culturally acceptable. Health promotion strategies aimed at improving a mother’s motivation to exclusively breastfeed may offer a cost-efficient and effective solution for mitigating childhood stunting in otherwise under-resourced global settings.

## Figures and Tables

**Figure 1 nutrients-13-04264-f001:**
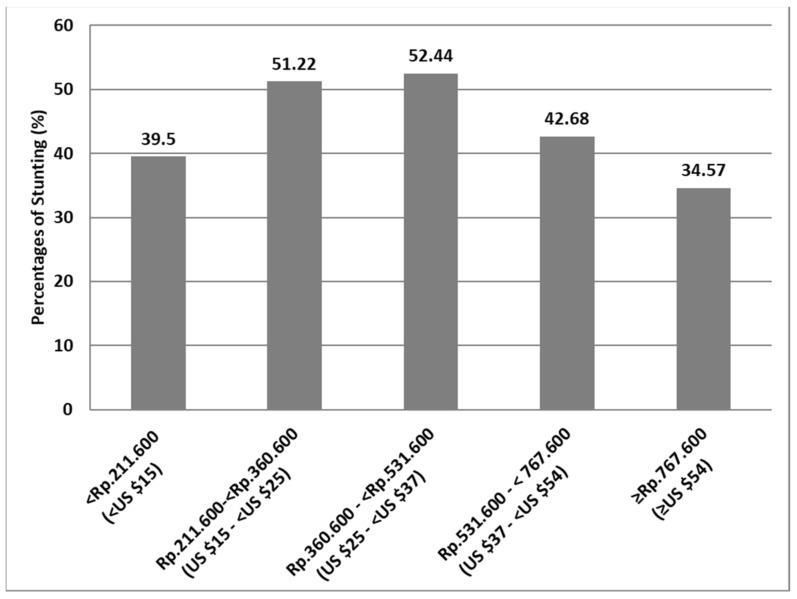
The Prevalence of Stunting according to Monthly Household Expenditures.

**Figure 2 nutrients-13-04264-f002:**
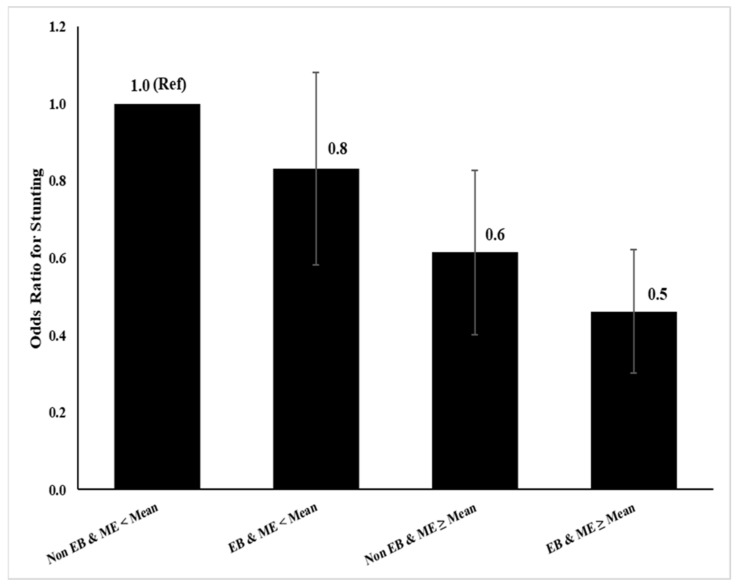
The Interaction Between Exclusive Breastfeeding (EB) & Monthly Expenditure (ME). We generated the Odds Ratio and 95% CI from multiple logistic regression after adjusting for sex, child age, child care, mother education, and father occupation.

**Table 1 nutrients-13-04264-t001:** Household Characteristics.

Characteristics	N	%
Number of Children		
1	114	27.9
2	92	22.5
3	80	19.6
4	59	14.5
5+	63	15.5
Father’s Age (years) Mean ± SD = 34.67 ± 8.7 Minimum = 19, Maximum = 76 Mother’s and Caregiver’s Age (years) Mean ± SD = 30.69 ± 8.0 Minimum = 17, Maximum = 72		
Categorized Mother’s Age		
<20 20–35	14 286	3.4 70.1
>35	108	26.5
Mother’s Education		
Did not finish elementary school	37	9.1
Graduated from elementary school	185	45.4
Graduated from junior high school	99	24.3
Graduated from senior high school	79	19.4
Graduated from university or equal	8	1.9
Father’s Education		
Did not finish elementary school	35	8.8
Graduated from elementary school	140	35
Graduated from junior high school	111	27.8
Graduated from senior high school	104	25.6
Graduated from university or equal	10	2.5
Mother’s Occupation		
Farmer/Breeder/Fisherman/Laborer	44	10.8
Civil Servant/Police/Military/Entrepreneur	9	2.2
Housewife/Unemployed	346	84.8
Other	9	2.2
Father’s Occupation		
Farmer/Breeder/Fisherman/Laborer	318	79.5
Civil Servant/Police/Military/Entrepreneur	67	16.8
Housewife/Unemployed	1	1.2
Other	2	2.5
Monthly Household Expenditure (mean ± SD)		
IDR 529.649 ± 385.522		
Monthly Household Expenditure Percentile		
<IDR 211,600 (USD 15)	81	19.8
IDR 211,600 - <IDR 360,600 (USD 15 - <USD 25)	82	20.1
IDR 360,600 - <IDR 531,600 (USD 25 - <USD 37)	82	20.1
IDR 531,600 - <IDR 767,600 (US $37 - <USD 54)	82	20.1
≥IDR 767,600 (USD 54)	81	19.9
Household Expenditure Based on Regional Minimum Wage		
<IDR 850.000 *	65	15.9
≥IDR 850.000	343	84.1

* Regional Minimum Wage for NTT Province in 2012 = IDR 850.000 (±USD 60.00).

**Table 2 nutrients-13-04264-t002:** Determinants of Stunting.

Determinants of Stunting	Stunting		Normal		COR * (95% CI)	AOR ^$^
	No	(%)	No	(%)		(95% CI)
Sex					1.52 (1.02–2.24) 1	
Boys	100	49.26	103	50.74		1.51 (0.98–2.33)
Girls	80	39.02	125	60.98		1
Exclusive Breastfeeding					0.82 (0.55–1.22) 1	
Yes	105	42.17	144	57.83		0.82 (0.52–1.30)
No	75	47.17	84	52.83		1
Age						
<12 months	35	22.44	121	77.56	0.21 (0.13–0.33) 1	0.20 (0.13–0.33)
12–24 months	145	57.54	107	42.46		1
Caregiver Type						
Mother	164	42.93	218	57.07	0.47 (0.18–1.14)	0.38 (0.14–1.03)
Other than Mother **	16	61.54	10	38.46	1	1
Caregiver’s Age						
<20 20–35 years	05 124	35.71 43.36	09 162	64.29 56.64	0.62 (0.15–2.23) 0.85 (0.53–1.36)	
>35 years	51	47.22	57	52.78	1	
Time Available to Look After Children						
Yes	174	44.16	220	55.84	1.05 (0.31–3.75)	
No	6	42.86	8	57.14	1	
Monthly Household Expenditure						
1st Quintile	32	39.5	49	60.5	1.24 (0.65–2.34)	1.03 (0.50–2.12)
2nd Quintile	42	51.2	40	48.8	2.00 (1.06–3.73)	2.00 (0.99–4.00)
3rd Quintile	43	52.4	39	47.6	2.09 (1.11–3.92)	2.28 (1.12–4.64)
4th Quintile	35	42.7	47	57.3	1.41 (0.75–2.66)	1.20 (0.60–2.41)
5th Quintile	28	34.6	53	65.4	1	1
Mother’s Education						
Uneducated	17	45.95	20	54.05	1.7 (0.77–3.73)	
Graduated Elementary School	85	45.95	100	54.05	1.7 (0.99–2.89)	
Graduated Junior High School	49	49.49	50	50.51	1.96 (1.08–3.55)	
Graduated Senior High School or Higher	29	33.33	58	66.67	1	
Mother’s Occupation						
Farmer/Breeder/Fisherman	22	51.16	21	48.84	9.43 (1.09–81.00)	11.74 (1.27–108.18)
Labor/Farming/Civil Servant/Military/ Entrepreneur	2	25	6	75	3.00 (0.22–40.93)	4.52 (0.29–69.86)
Unemployed	155	44.67	192	55.33	7.27 (0.91–57.97)	8.40 (1.00–70.29)
Other	1	10	9	90	1	1
Father’s Age						
25–39 years	110	42.31	150	57.69	0.85 (0.54–1.30)	
<25 & ≥40 years	65	46.43	75	53.57	1	
Father’s Occupation						
Farmer/Breeder/Fisherman	137	46.13	160	53.87	0.96 (0.53–1.72)	
Labor/Farming/Civil servant/Military/ Entrepreneur	15	28.30	38	71.70	0.44 (0.20–0.99)	
Unemployed	3	60.00	2	40	1.68 (0.26–10.88)	
Other	20	44.44	25	55.56	1	
Monthly Household Expenditure Based on Regional Minimum Wage						
<IDR 850.000 (USD 60.00) ≥IDR 850.000 (USD 60.00)	185 43	53.94 66.15	298 22	46.06 33.85	1.66 (0.96–2.91) 1	

* COR or Crude Odds Ratio and 95% Confidence Interval were generated from a simple logistic regression. ^$^ AOR or Adjusted Odds Ratio and 95% Confidence Interval were generated from multiple logistic regression model adjusting for the remaining variables. ** Other than Mother in caregiver variable means that because of certain conditions, some children were not cared by their biological mother, but cared by their father, grandmother, aunt, sibling, or other family member.

**Table 3 nutrients-13-04264-t003:** Bivariate Analysis Between Monthly Household Expenditures and Exclusive Breastfeeding.

Variable	Exclusive Breastfeeding		Not Exclusive Breastfeeding		*X*^2^ * (*p*-Value)
	No.	(%)	No.	(%)	
Monthly Household Expenditures					
<IDR 211,600 (USD 15)	57	70.37	24	29.63	11.9 (0.018)
IDR 211,600 - <IDR 360,600 (USD 15 - <USD 25)	50	60.98	32	39.02	
IDR 360,600 - <IDR 531,600 (USD 25 - <USD 37)	57	69.51	25	30.49	
IDR 531,600 - <IDR 767,600 (USD 37 - <USD 54)	46	56.10	36	43.90	
>IDR 767,600 (USD 54)	39	48.15	42	51.85	
Total	249	61.03	159	38.97	

* Chi-Square test.

## Data Availability

The data are not publicly available due to participant confidentiality.

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
