# Peer review of "Exclusive Breastfeeding Protects Young Children from Stunting in a Low-Income Population: A Study from Eastern Indonesia"

_nutrients, 2021, doi:10.3390/nu13124264_

Round 1

Reviewer 1 Report

This article deals with the important issue of breastfeeding and promoting baby feeding as recommended by the WHO. It is of particular importance in countries suffering from poverty and food safety problems.

  1. In the abstract, the authors indicate that the study covered children aged 6 to 24 months, while in the chapter "Materials and methods" (2.1. Study Population and Sampling Methods) they indicate that children aged 6 months to 23 months were examined. Please clarify or correct the text.
  2. In the chapter "Materials and methods" (2.3.4. Birth order) the authors explain that they divided the studied children into two categories in terms of the order of birth in the family: "i) child below or at number 2 or ii) child above number 2". In contrast, in the chapter "results" we see the presented results and the division of children in the order of birth as the first, second, third in the family.I think that it is not worth using the previous classification given in the "Material and methods" section, because the obtained results allow for a more detailed presentation of the results, as the authors did.
  3. In the "Ethical Approvals" section, it is helpful to include the number or tag of the Ethics Committee approval obtained, if available.
  4. Please verify the correctness of the given maximum age of the father and mother participating in the study. I am not saying that there is a mistake, but for the region of the world I come from, it is a high age to be a parent of such young children. That's why it seems unusual to me.
  5. It would be useful to include data on the views and knowledge about breastfeeding of the parents participating in the research. I think you have statistics on the parents' knowledge. The authors presented this information more descriptively without providing specific values.
  6. In Table 2 in the line headed "Caregiver Type", the percentages have been mixed up: Caregiver - Mother stunding No 164 is 42.93% (not 61.54% as it is written). Similarly, Caregiver Other than Mother 16 is 61.54% (not 42.93% as it is written). These values are incorrectly converted with each other.

Reviewer 2 Report

The authors report on an interesting and important topic of childhood stunting and the potential of exclusive breastfeeding to reduce this.

I have some suggestions for improvement of the manuscript:

  • The abstract/intro mentions that breastfed infants grow better than non-breastfed. However, formula-fed children often grow very well so this might need to be addressed for LMIC specificly. 
  • Is there any further information available about breastfeeding practise? The authors mention the importance of colostrum intake. Is there data on this practise in the cohort?
  • Paragraph 3.2 mentions caregiver type as an important factor for stunting rates. Is this data significant as the 95% CI includes 1? The data in the table needs more careful analysis and description in the text. 
  • Line 220-223 states 'In addition, children from households with a monthly income of less than the regional minimum wage were more likely to be stunted compared to those from households with a monthly income of greater than or equal to the regional minimum  wage (Table 2).' This would be very interesting data to include, but cannot be extracted from Table 2 as it is. Please include. 
  • Line 253: should 60% read 40%?
  • It would be interesting to include data/statistics on substitute feeds given to see if there is associations between the type of feeds infants receive and stunting. Currently very limited info is provided.
  • The data in Figure 1 is already available in Table 1. There is no need to duplicate the data (income brackets can be added to the table).
  • It would be good to include breastfeeding data for every income quintile (currenly this is briefly covered in the discussion, but this should be addressed in the results section in more detail). 
  • From line 259 onwards a large part of the text belongs in the discussion. Please reformat this. 

Reviewer 3 Report

The paper has several gaps, in many parts. Both at the content level and at the structural level. I recommend a thorough review also according to the following suggestions.
Point 1: In the abstact there is no mention of when the interviews were carried out. Furthermore, there is no mention of the methodology used at the data processing level. Above all, the contribution that this paper has to the scientific literature is not mentioned. Does it update existing data, or is it a new study for the topic? If there are other similar studies, what does it bring on a scientific level?
Point 2: from the introductory point of view, several reports rich in statistical data are inserted. Why are two paragraphs not made to differentiate the already existing studies (literature review) and one where these statistical data are reported? It would be clearer to the reader. Did the authors ask themselves any research questions or research hypotheses? If not present it is recommended to add them.
Point 3: There is no paragraph on the study area. The methodology immediately speaks of the subjects interviewed. The subjects, is 408 a significant figure that represents the entire sample? Was the survey carried out before or after the pandemic - there is no reference to the survey year. There are too many subsections within the materials and methods. try to organize better by eliminating some and making the text fluid. There can be no subparagraphs of 3 lines, if anything make a table where the variables described are inserted, such as categories, age and education.
Expand the part on data analysis. Make references also at a bibliographic level.
Point 4: In the discussions section, answer the research questions that it was suggested to insert earlier. The conclusions are too short, expand them.
Point 5: in the references Articles partly not updated. Integrate with other recent references.

Round 2

Reviewer 3 Report

The authors modified the paper according to the suggestions proposed in the first round of review in an excellent way. I believe the article may now be publishable